# Bioactive Compounds from the Mushroom-Forming Fungus *Chlorophyllum molybdites*

**DOI:** 10.3390/antibiotics12030596

**Published:** 2023-03-16

**Authors:** Jing Wu, Takeru Ohura, Ryuhei Ogura, Junhong Wang, Jae-Hoon Choi, Hajime Kobori, Corina N. D’Alessandro-Gabazza, Masaaki Toda, Taro Yasuma, Esteban C. Gabazza, Yuichi Takikawa, Hirofumi Hirai, Hirokazu Kawagishi

**Affiliations:** 1Faculty of Agriculture, Shizuoka University, 836 Ohya, Suruga-ku, Shizuoka 422-8529, Japan; wu.jing@shizuoka.ac.jp (J.W.);; 2Research Institute for Mushroom Science, Shizuoka University, 836 Ohya, Suruga-ku, Shizuoka 422-8529, Japan; 3Graduate School of Integrated Science and Technology, Shizuoka University, 836 Ohya, Suruga-ku, Shizuoka 422-8529, Japan; 4Graduate School of Science and Technology, Shizuoka University, 836 Ohya, Suruga-ku, Shizuoka 422-8529, Japan; 5Research Institute of Green Science and Technology, Shizuoka University, 836 Ohya, Suruga-ku, Shizuoka 422-8529, Japan; 6Iwade Research Institute of Mycology Co., Ltd., Suehirocho 1-9, Tsu 514-0012, Japan; 7Department of Immunology, Mie University Graduate School of Medicine, Edobashi 2-174, Tsu 524-8507, Japan

**Keywords:** mushroom, *Chlorophyllum molybdites*, structure determination, Axl inhibitor, immune checkpoint inhibitor, anti-phytopathogenic-bacterial activity, plant growth regulator

## Abstract

A novel compound (**1**) along with two known compounds (**2** and **3**) were isolated from the culture broth of *Chlorophyllum molybdites*, and three known compounds (**4**–**6**) were isolated from its fruiting bodies. The planar structure of **1** was determined by the interpretation of spectroscopic data. By comparing the specific rotation of the compound with that of the analog compound, the absolute configuration of **1** was determined to be *R*. This is the first time that compounds **2**–**4** were isolated from a mushroom-forming fungus. Compound **2** showed significant inhibition activity against Axl and immune checkpoints (PD-L1, PD-L2). In the bioassay to examine growth inhibitory activity against the phytopathogenic bacteria *Peptobacterium carotovorum*, *Clavibacter michiganensis* and *Burkholderia glumae*, compounds **2** and **3** inhibited the growth of *P. carotovorum* and *C. michiganensis.* In the bioassay to examine plant growth regulatory activity, compounds **1**–**4** showed a significant regulatory activity on lettuce growth.

## 1. Introduction

The fruiting body of certain kinds of eukaryotic, non-photosynthetic and aerobic fungi is generally known as a mushroom. The mushroom-forming fungi produce spores, and the spores germinate and create mycelia. The mycelia eventually produce primordia, which grow into new whole mushrooms, and the life cycle continues. Based on the taxonomic classification, the mushroom-forming fungi are divided into two groups, Basidiomycetes and Ascomycetes [1,2]. There is an expression that says, “plants act as producers, animals as consumers, and fungi as restorers or decomposers”. In other words, plants create organic substances (carbohydrates) by photosynthesis and animals consume such plants. Then fungi, including mushroom-forming ones, play important roles in bringing the plants and animals back to the land. There are large differences in the structures of metabolites produced by mushroom-forming fungi compared to those produced by plants and animals, and biological activities indigenous to mushroom-forming fungi are often due to the differences [3,4]. Mushroom extracts and their secondary metabolites have been found to have various biological activities such as antioxidant, antimicrobial, antiobesity and immunomodulatory activities. The value of medicinal properties is increased, especially for their activities on cardiometabolic parameters, on the immune system and as anti-inflammatory and anticancer agents [5,6,7]. We also found nerve growth factor (NGF) stimulators from the fruiting bodies and mycelia of *Hericium erinaceus*, and named them hericenones C to H and erinacines A to I, which stimulate the biosynthesis of NGF and are considered to be effective against dementia [8]. There is a growing interest in the medicinal use of nutritional products derived from mushrooms. Mushroom extracts are increasingly available as dietary supplements, especially to increase immune function and anticancer activities [1].

Cancer is one of the leading causes of death worldwide, and according to the WHO, it accounted for nearly 10 million deaths in 2020 [9]. There are many reports demonstrating the beneficial effects of mushrooms on cancer treatment [10]. The polysaccharides (PLP) isolated from *Phellinus linteus* inhibit tumor growth and lung metastasis by stimulating the immune response and have no direct toxic effect on cancer cells [11]. Triterpenoids from *Ganoderma lucidum* shows anticancer properties [12]. *β*-d-glucans in *G. lucidum* shows anti-cancer effects by inhibiting cancer cells, protecting normal cells from free radicals and reducing damage to normal cells [13]. We also have reported the isolation and structure determination of an anti-tumor *β*-(1→6)-d-glucan-protein complex from the fruiting bodies of *A. blazei* [14].

*Chlorophyllum molybdites* belongs to the family Agaricaceae, which is a kind of poisonous mushroom, and is found in western Japan and the Tokai region throughout the rainy season in Japan. The fruiting bodies of this fungus have various biological properties, such as antimicrobial and antiplasmodial activities [15,16]. Chemical study of the fruiting bodies of this mushroom led to the isolation of anti-cancer steroids, pyrrolidine derivative alkaloid lepiotin B and a toxic protein molybdophyllysin [17,18,19]. We also have reported previously the purification and characterization of an *N*-glycolylneuraminic acid-specific lectin from the fruiting bodies [20].

During screening, we also found the receptor tyrosine kinase (Axl) and immune checkpoint inhibitory activities and anti-phytopathogenic-bacterial activity of extracts of the fruiting bodies of *C. molybdites*, and we attempted to find the active compounds from this mushroom.

As one of the most common and serious kinds of cancer, lung cancer is the leading cause of cancer death all over the world [21]. Axl and programmed death ligands 1 and 2 (PD-L1 and PD-L2) have been intensively studied in cancer treatment [22,23,24]. Activation of Axl signaling stimulates cell survival and increases the migration and invasion of cancer cells. The Axl pathway also enhances immune evasion in the tumor microenvironment cells. As a key factor in drug resistance and metastasis, Axl has been extensively implicated in the epithelial–mesenchymal plasticity of cancer cells [25,26,27,28]. On the other hand, programmed cell death-1 (PD-1) is activated upon binding to its ligands PD-L1 and PD-L2, which is an important inhibitory receptor expressed on the surface of activated T cells and B cells [29,30]. In cancer cells, the expression of PD-L1 and PD-L2 is a very important part of the mechanism contributing to the immune escape of cancer cell [31]. Several antibody-based or small-molecule Axl, PD-L1 and PD-L2 inhibitors have been developed and used in preclinical studies [29,32,33,34]. Recently, we have focused on Axl, PD-L1 and PD-L2 inhibitors and found their inhibitors from the mushrooms *Leucopaxillus giganteus, Pleurocybella porrigens* and *Lepista luscina* [35].

Plant diseases are one of the major causes of global total crop production losses. The severity of disease outbreaks caused by plant phytopathogenic fungi and bacteria has been steadily increasing over the past decades [36]. For example, fungal leaf spot of maize cause by *Drechslera maydis* in warm humid areas and leaf rust caused by *Melampsora* spp. in willow plantations are the most serious and destructive diseases [37,38]. Pesticides including bactericides and fungicides play an important role in plant disease management. However, pesticide residues pose a serious threat to environmental, biodiversity and human health due to their slow biodegradation [39]. Therefore, it is important to search for effective chemicals from natural sources to suppress phytopathogenic bacteria without environmental pollution. Recently, we found that anti-phytopathogenic-bacterial fatty acids were isolated from the mushrooms *A. blazei* [40].

On the other hand, *C. molybdites* is known to form fairy rings, and normally found in farmlands, lawns, etc. It is suitable for artificial cultivation [15]. Fairy rings are an interesting phenomenon in which the growth of grass is promoted and/or inhibited by the interaction between fungi and plants worldwide [41]. We discovered three plant growth regulators, 2-azahypoxanthine (AHX), 2-aza-8-oxohypoxanthine (AOH) and imidazole-4-carboxamide (ICA), as the fairy-ring-causing principles. Our study of fairy rings was covered in *Nature* and we named them “fairy chemicals” after the title of the article in the journal [42]. AHX and ICA were found from the culture broth of the fairy-ring-forming fungus *Lepista sordida,* and AOH was isolated from AHX-treated rice as a metabolite of AHX [43]. Recently, we also reported that AHX is a promising anti-angiogenic agent in retinal neovascularization by inhibiting the activation of hypoxia inducible factor [44]. AOH is effective as a cosmetic ingredient with a skin barrier function against water loss and skin lightening [45]. ICA inhibited the expression of Axl receptor tyrosine kinase and immune checkpoint molecules in melanoma cells *in vitro* and improved the therapeutic response to cisplatin in mouse melanoma xenografts *in vivo* [46]. Additionally, erinaceolactones A and B, erinachromanes A and B and erinaphenol A were isolated as plant growth inhibitors from the culture broth of *Hericium erinaceus* [2]. Plant growth regulators, armillariols A to C and sesquiterpene aryl esters, were isolated from the culture broth of *Armillaria* sp. They might play some roles in the Armillaria root disease [2]. We continue to search for the substances that interact between plants and fungi in *C. molybdites.*

Therefore, we attempted to find Axl and immune checkpoint inhibitors, anti-phytopathogenic-bacterial and plant-growth-regulating compounds from both the culture broth and fruiting bodies of *C. molybdites*. As a result, a novel compound (**1**) along with two known compounds (**2** and **3**) were isolated from the culture broth, and three known compounds (**4**–**6**) were isolated from the fruiting bodies. Here, we describe the isolation, structure determination and Axl immune checkpoint inhibitory activities of the compounds. In addition, we report anti-phytopathogenic-bacterial activity and plant growth regulation activity of these compounds.

## 2. Results and Discussion

The culture broth of *C. molybdites* was partitioned between *n*-hexane and water, and then EtOAc and water, successively. The EtOAc-soluble parts were fractionated with repeated chromatography, and a novel compound (**1**) and two known compounds (**2** and **3**) were isolated (Figure 1). The fresh fruiting bodies of *C. molybdites* were extracted with EtOH and then with acetone. After the solutions were combined and concentrated, they were partitioned between *n*-hexane and water, EtOAc and water, and the water part concentrated under reduced pressure, and then extracted with EtOH, successively. The EtOAc-soluble part and the EtOH-soluble part were fractionated with repeated chromatography. As a result, three known compounds (**4**–**6**) were isolated (Figure 1).

Compound **1** was obtained as brown amorphous. The molecular formula was determined as C_10_H_11_NO_5_ with HRESIMS (*m/z* 224.0584 [M-H]^−^; calcd for C_10_H_10_NO_5_, 224.0565), indicating the presence of six degrees of unsaturation in the molecule. The structure of **1** was elucidated through the interpretation of NMR spectra including DEPT, HMQC, COSY and HMBC (Appendix A). The DEPT experiment indicated the presence of one methylene, five methines and four tetrasubstituted carbons including two carboxy groups (*δ*_C_ 171.0, 173.6). The complete assignment of all the protons and carbons was accomplished as shown in Table 1. The 4-aminobenzoic acid group was constructed based on the characteristic ^1^H and ^13^C NMR chemical shifts and coupling constants [1-COOH, *δ*_C_ 171.0; C-1, *δ*_C_ 129.5; C-2, 6, *δ*_H_ 7.96 (d, 8.4), *δ*_C_ 131.6; C-3, 5, *δ*_H_ 7.71 (d, 8.4), *δ*_C_ 120.4; C-4, *δ*_C_ 142.7), the HMBC correlations (H-2/C-3, C-4, C-6, 1-COOH; H-3/C-1, C-4, C-5) and the COSY correlations (H-2/H-3, H-5/H-6) (Figure 2). The ^1^H and ^13^C NMR chemical shifts [C-1′, *δ*_C_ 173.6; C-2′, *δ*_H_ 4.20 (dd, 4.6, 3.7), *δ*_C_ 74.6; C-3′, *δ*_H_ 3.80 (dd, 11.6, 4.6), 3.83 (dd, 11.6, 3.7), *δ*_C_ 65.3), the HMBC correlations (H-2′/C-1′, H-3′/C-1′), the COSY correlations (H-2′/H-3′) and the molecular formula suggested the presence of the 2,3-dihydroxypropanoate moiety and the amide bond (Figure 2). Hence, its planar structure was determined to be 4-(2,3-dihydroxypropanamido)benzoic acid. To determine the absolute configuration of **1**, the specific rotation {[α]_D_^23^ +29 (*c* = 0.12, MeOH)} was compared with that of its analog, (*R*)-2,3-dihydroxy-*N*-(4-vinylphenyl)propenamide {[α]_D_^20^ +59.2 (*c* = 1.30, acetone)}, whose absolute configuration has been determined [47]. All the data allowed us to conclude that compound **1** was a novel compound, (*R*)-4-(2,3-dihydroxypropanamido)benzoic acid (Figure 1).

Compounds **2** and **3** were identified as fusaric acid and 9,10-dehydrofusaric acid by the comparison of their spectroscopic data with those reported [48]. Compounds **2** and **3** have been isolated from the culture filtrate of *Fusarium nygamai*, which showed a strong inhibition of root elongation of seedlings as well as wide chlorosis of tomato leaves rapidly evolving into necrosis [48]. Both the compounds were first isolated from mushroom-forming fungi. Compound **4** was identified as ethyl 2-acetylamino-2-deoxy-*β*-d-glucopyranoside, which was isolated from *Aspergillus terreus* and has growth-promoting activity for *Lactobacillus bifidus* var. *pennsylvanicus* [49]. This compound also was isolated from mushrooms for the first time. Compound **5**, 4-ethoxy-4-oxobutanoic acid, has been isolated from the mushroom *Trametes versicolor* [50], which has insulinotropic action in rat islets [51]. Compound **6** was identified as methyl 4-hydroxyphenylacetate, which was isolated from the fungus *Gloeophyllum odoratum* and a marine fungus *Penicillium oxalicum*, and has potent inhibitory activity against tobacco mosaic virus [52,53].

The human A549 alveolar epithelial cell lines were treated with compounds **2** and **3**. As shown in Figure 3, compound **3** showed no effects on all the gene expressions. However, compound **2** significantly inhibited the expression of Axl, PD-L1 and PD-L2. The difference in the activity between **2** and **3** indicated that a butyl group played an important role in the suppression of all three genes.

We also examined effect of compounds **2** and **3** on the growth of *Pectobacterium carotovorum*, *Clavibacter michiganensis* and *Burkholderia glumae*. *Pectobacterium* species are economically important plant pathogens and cause soft rot and blackleg disease on a range of plant species around the world [54,55]. Among the *Pectobacterium* species, *P. carotovorum* is a Gram-negative plant-specific pathogen and has the widest host range that causes soft rot disease in diverse plants [56]. Potato is the most important crop affected in temperate regions [54,57]. A Gram-positive plant pathogenic bacterium, *C. michiganensis*, is one of the most disruptive bacterial diseases of tomato [58]. This bacterium gives a serious threat to the processing and fresh market tomato industries and causes catastrophic epidemics in most tomato-growing areas of the world. In general, this vascular pathogen causes wilt and canker symptoms by invading and diffusing in the xylem through natural openings or wounds [59,60]. *B. glumae,* as a Gram-negative bacterium, was first described in Japan leading to grain rotting and seedling blight on rice [61,62], and is an emerging rice disease that greatly limits the productivity of rice [63]. We used ampicillin as a positive control that has anti-phytopathogenic-bacterial activity. As a result, compounds **2** and **3** inhibited the growth of *P. carotovorum* and *C. michiganensis* at 0.1 μmol/paper disc, while showing no activity against the growth of *B. glumae* (Figure 4).

In addition, the effect of compounds **1**–**6** and the analog compounds of **4** and **6** [*N*-acelyl-d-glucosamine and methyl 2-(3-hydroxyphenyl)acetate] on lettuce growth was evaluated (Figure 5). We used 2,4-dichlorophenoxyacetic acid (2,4-D) as a positive control, which inhibited the root and hypocotyl growth of lettuce dose-dependently. Compound **1** weakly promoted the root growth at 1 μmol/paper, while it inhibited the hypocotyl growth of lettuce at 1, 10 and 100 nmol/paper (Figure 5A). As for the root and hypocotyl growth of lettuce, compounds **2**–**4** and methyl 2-(3-hydroxyphenyl)acetate showed inhibition activity at 100 and 1000 nmol/paper, respectively(Figure 5B,C). Compounds **5**, **6** and *N*-acelyl-d-glucosamine exhibited no activity (Figure 5C). Interestingly, a comparison of the structures between **4** and *N*-acelyl-d-glucosamine indicated that the ethoxy group at C-2 strengthened the lettuce-growth-inhibitory activity. The comparison between **6** and methyl 2-(3-hydroxyphenyl)acetate suggested that the hydroxy group at the meta position played an important role in the stronger inhibitory activity than at the para position.

## 3. Materials and Methods

### 3.1. General Experimental Procedures

^1^H NMR spectra (one- and two-dimensional) were recorded on a Jeol lambda-500 spectrometer or a JNM-ECZ500R spectrometer at 500 MHz, and ^13^C NMR spectra were recorded on the same instrument at 125 MHz (JEOL, Tokyo, Japan). HRESIMS spectra were measured on a JMS-T100LP mass spectrometer (JEOL, Tokyo, Japan). The specific rotation values were measured with a Jasco DIP-1000 polarimeter (Jasco, Tokyo, Japan). HPLC separations were performed with a Jasco Chromatography Data Station ChromNAV system using reverse-phase HPLC columns (CAPCELL PAK C18 AQ, Osaka soda, Osaka, Japan; COSMOSIL PBr, nacalai tesque, Kyoto, Japan; InerSustain Amide, InertSustain Phenyl, GL Science, Tokyo, Japan). Silica gel plate (Merck F254), ODS gel plate (Merck F254) and silica gel 60 N (Kanto Chemical, Tokyo, Japan) were used for analytical TLC and for flash column chromatography. All solvents used throughout the experiments were obtained from Kanto Chemical Co. (Tokyo, Japan).

### 3.2. Fungal Material

Fresh fruiting bodies of *C. molybdites* were collected at Tsu, Mie Prefecture, Japan, in 2015. The culture mycelium was isolated from the fruiting bodies successfully and then identified as *C. molybdites* by determining the internal transcribed spacer (ITS) regions of nuclear ribosomal DNA (rDNA) sequences deposited at NCBI BLAST (http://blast.ncbi.nlm.nih.gov/). The mycelia of *C. molybdites* were pre-cultured on potato dextrose agar (PDA), and the inoculated mycelia were incubated at 25°C for two weeks. After growth, 10 pieces (6 mm diameter) cut from the two-week-cultured mycelia were inoculated into 500 mL Erlenmeyer flasks containing 300 mL of PDB medium (*n* = 5), and the cultures were incubated for 4 weeks (25 °C, 120 rpm). Lettuce seeds (*Lactuca sativa* L. cv. Cisko; Takii Co., Ltd., Tokyo, Japan) were used in this study.

### 3.3. Extraction and Isolation

The culture broth of *C. molybdites* (1.5 L) was filtered and then concentrated under reduced pressure. The concentrate was successively partitioned between *n*-hexane and water (2 L each, twice), and then ethyl acetate (EtOAc) and water (2 L each, twice), to obtain the *n*-hexane-soluble part, EtOAc-soluble part, and water-soluble part. The EtOAc-soluble part (111.3 mg) was separated with Sep-Pak ODS (30% MeOH and MeOH). The 30% MeOH elution part (72.4 mg) was further fractionated using reverse-phase HPLC (COSMOSIL PBr, 20% MeOH) to yield 12 fractions (fractions CB-EtOAc-30MeOH-1 to CB-EtOAc-30MeOH-12). Compound **1** (1.2 mg) was isolated from fraction CB-EtOAc-30MeOH-9 (5.7 mg) with reverse-phase HPLC (CAPCELL PAK C18 AQ, 20%MeOH). The water-soluble part (2.8 g) was further extracted with EtOAc three times. The water-EtOAc-soluble part (215.7 mg) was separated with Sep-Pak ODS (30% MeOH and MeOH). Fraction CB-H_2_O-EtOAc-30MeOH (193.4 mg) was further fractionated using reverse-phase HPLC (CAPCELL PAK C18 AQ, 30% MeOH + 0.05%TFA) to yield 9 fractions (fractions CB-H_2_O-EtOAc-30MeOH-1 to CB-H_2_O-EtOAc-30MeOH-9). Fraction CB-H_2_O-EtOAc-30MeOH-7 was compound **2** (22.2 mg), and fraction CB-H_2_O-EtOAc-30MeOH-5 (26.3 mg) was further fractionated using reverse-phase HPLC (InertSustain Phenyl, 70% MeOH) to obtain compound **3** (15.6 mg).

The fresh fruiting bodies of *C. molybdites* (1.25 kg) were extracted with EtOH (5 L each, twice) and then with acetone (5 L each, twice). After the solutions were combined and concentrated under reduced pressure, the concentrate was partitioned between *n*-hexane and water, EtOAc and water, and the water part concentrated under reduced pressure, and then extracted with EtOH successively. The EtOH-soluble part (2.1 g) was fractionated using silica gel flash column chromatography (CH_2_Cl_2_, 70%, 50%, 30%, 10% CH_2_Cl_2_/MeOH; MeOH) to obtain 11 fractions (Fractions FB-EtOH-1 to FB-EtOH-11). Fraction FB-EtOH-5 (227.2 mg) was further fractionated with reverse-phase HPLC (CAPCELL PAK C18 AQ, 15% MeOH) to yield 10 fractions (Fractions FB-EtOH-5-1 to FB-EtOH-5-10). Compound **4** (0.8 mg) was isolated from fraction FB-EtOAc-5-3 (10.3 mg) with reverse-phase HPLC (InerSustain Amide, 95% MeCN). The EtOAc-soluble part (3.7 g) was fractionated using silica gel flash column chromatography (CH_2_Cl_2_, 90%, 70%, 50%, 30%, 10% CH_2_Cl_2_/acetone; acetone; 70%, 50%, 30%, 10% acetone/MeOH; MeOH) to obtain 13 fractions (Fractions FB-EtOAc-1 to FB-EtOAc-13). Fraction FB-EtOAc-6 (90.5 mg) was further fractionated with reverse-phase HPLC (COSMOSIL PBr, 60% MeOH) to obtain compound **5** (1.0 mg). Fraction FB-EtOAc-8 (80.8 mg) was further fractionated with reverse-phase HPLC (COSMOSIL PBr, 60% MeOH) to obtain compound **6** (1.0 mg).

### 3.4. Axl and Immune Checkpoint Molecule Assay [35]

The human A549 alveolar epithelial cell line was purchased from the American Type Culture Collection (Rockville, MD, USA) and cultured in DMEM, supplemented with 10% heat-inactivated fetal bovine serum, 2 mM L-glutamine and 100 U mL penicillin plus 100 μg/mL streptomycin. All cells were cultured at 37 °C in 75 cm^2^ flasks in an atmosphere composed of 5% CO_2_ and 95% air. Confluent cells were passaged after 5–7 days.

A549 cells in 0.1% BSA-DMEM were seeded in 24-well plates. Test compounds (20 μg/mL) were added to the wells, and the plates were incubated for 24 h. The total RNA was extracted using Sepasol®-RNA I Super G (Nacalai) following the instructions of the manufacturer. One μg of total RNA was denatured at 65°C for 10 min and then reverse-transcribed using ReverTra Ace Reverse Transcriptase (TOYOBO) and oligo (dT) primer in a volume of 20 μL according to the manufacturer’s protocol.

Each gene contains forward and reverse sequences (5′ > 3′), which are, respectively, GGAGCGAGATCCCTCCAAAAT and GGCTGTTGTCATACTTCTCATGG for the GADPH gene, TGCCATTGAGAGTCTAGCTGAC and TTAGCTCCCAGCACCGCGAC for the Axl gene, GGACAAGCAGTGACCATCAAG and CCCAGAATTACCAAGTGAGTCCT for the PD-L1 gene, and ACCGTGAAAGAGCCACTTTG and GCGACCCCATAGATGATTATGC for the PD-L2 gene. The cDNA was amplified using PCR and the conditions were as follows: 94 °C, 1 min; 60 °C, 1 min; and 72 °C, 1 min for 28–35 cycles. The PCR products were electrophoresed on a 1.5% agarose gel and then stained with an ethidium bromide solution. The semi-quantitative RT-PCR results were quantified using ImageJ software.

The data are expressed as the mean ± standard error of the mean (SEM). The statistical difference was calculated using analysis of variance with post hoc analysis using Fisher’s predicted least significant difference test. All statistics were performed using the StatView 5.0 package (Abacus Concepts, Berkeley, CA, USA).

### 3.5. Antibacterial Assay [40]

Each bacterium (*C. michiganensis*, *B. glumae* and *P. carotovorum*) was taken from the slant using an inoculation loop and suspended in 1 mL of sterile water in a 1.5 mL Eppendorf tube, and a suspension of 10^8^ colony forming units (CFU)/mL was made with reference to the OD_600_. YP medium (yeast extract 5 g/L, peptone 10 g/L, agar 15 g/L) in a test tube was autoclaved for 20 min at 121 °C. The medium was left to stand until the temperature reached about 30 °C, and 100 μL of each bacterium suspension was added to the medium, and the mixture was poured into a Petri dish.

A total of 40 μL of a solution of each compound (0.1, 0.05 and 0.01 μmol in MeOH) and MeOH only (as control) were put on a paper disc (8 mm in diameter). After the discs were dried in the air, they were put on the medium. They were incubated for 3 days to evaluate their antibacterial activity.

### 3.6. Plant-Growth-Regulating Assay

Lettuce seeds were put on filter paper (Advantec No. 2, *ϕ* 55 mm; Toyo Roshi Kaisha, Japan), soaked in distilled water in a Petri dish (*ϕ* 60 × 20 mm) and incubated in a growth chamber in the dark at 25 °C for 1 day. Each sample was dissolved in 1 mL of MeOH (1, 10, 100 and 1000 nmol/mL) and then poured on filter paper (*ϕ* 55 mm) in a Petri dish (*ϕ* 60 × 20 mm). After the sample-loaded paper had been air-dried, 1 mL of distilled water was poured on the sample-loaded paper or intact filter paper (control). The preincubated lettuce seedlings (*n* = 9 in each Petri dish) were transferred onto the sample-loaded filter paper or control filter paper and incubated in a growth chamber in the dark at 25 °C for 3 days. The lengths of the hypocotyl and the root were measured using a ruler.

## 4. Conclusions

A novel compound (**1**) along with two known compounds (**2** and **3**) were isolated from the culture broth of *C. molybdites*, and three known compounds (**4**–**6**) were isolated form the fruiting bodies of the fungus. Compounds **2**–**4** were isolated from a mushroom for the first time. Compound **2** showed significant inhibition activity against Axl and immune checkpoint genes (PD-L1, PD-L2) that are the most promising candidates for cancer therapy. In addition, compounds **2** and **3** inhibited the growth of *P. carotovorum* and *C. michiganensis*. Compounds **1**–**4** showed the significant regulatory activity of lettuce growth (Figure 6).

## Figures and Tables

**Figure 1 antibiotics-12-00596-f001:**
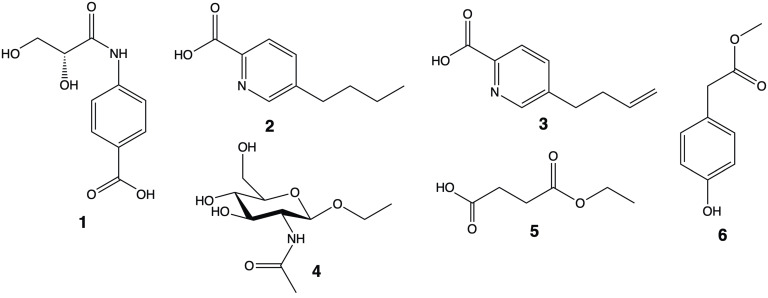
Structures of compounds **1**–**6**.

**Figure 2 antibiotics-12-00596-f002:**
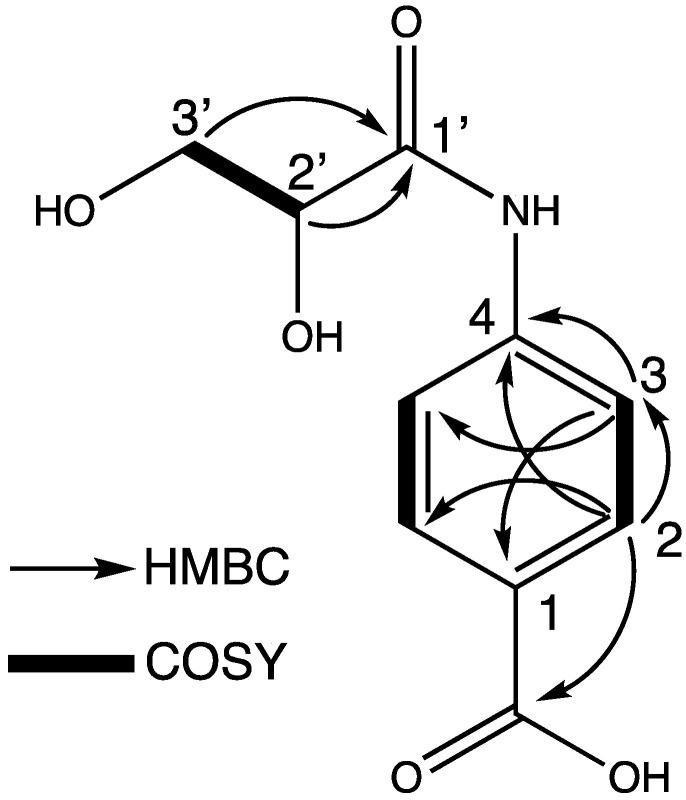
HMBC and COSY correlations of **1**.

**Figure 3 antibiotics-12-00596-f003:**
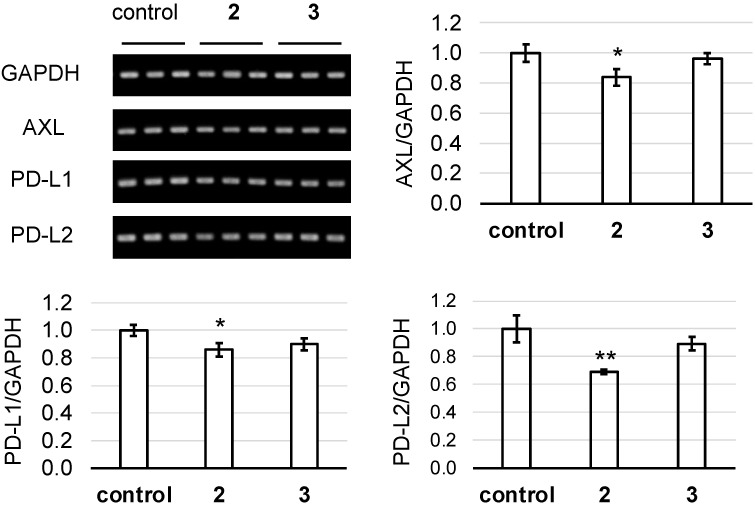
Effect of compounds **2** and **3** on expression of Axl and immune checkpoint molecules (PD-L1 and PD-L2) in A549 cells. Values indicate means with standard deviation. Statistical analysis was performed using Fisher’s PLSD test (* *p* < 0.05, ** *p* < 0.01 vs. control, *n* = 3).

**Figure 4 antibiotics-12-00596-f004:**
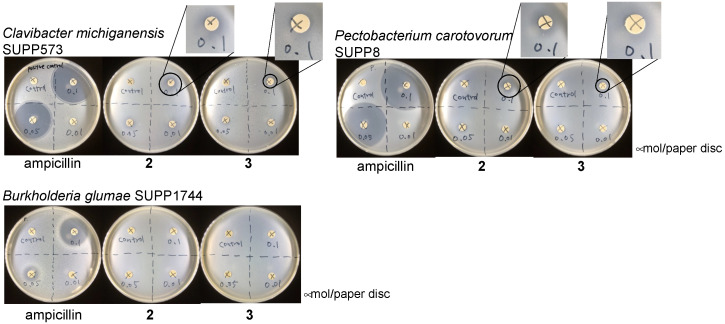
Activity of **2** and **3** against *Clavibacter michiganensis*, *Pectobacterium carotovorum* and *Burkholderia glumae* (positive control, ampicillin).

**Figure 5 antibiotics-12-00596-f005:**
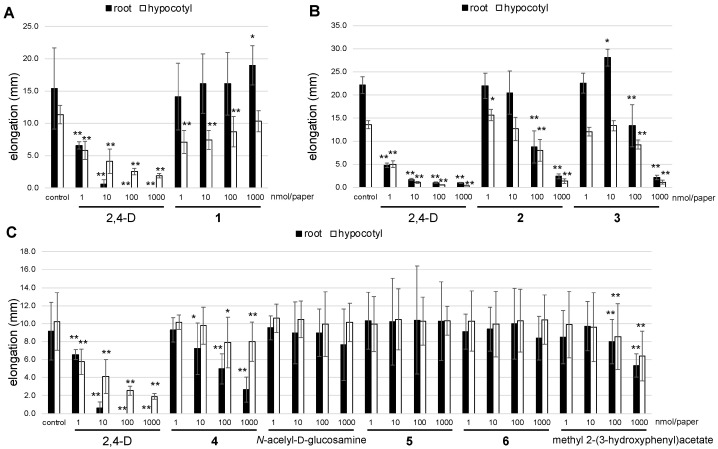
Effect of **1** to **6**, *N*-acelyl-d-glucosamine and methyl 2-(3-hydroxyphenyl)acetate on the growth of lettuce. (**A**). compound **1**; (**B**). compounds **2** and **3**; (**C**). compounds **4**–**6** and their analog compounds. Lettuce seedlings were treated with compounds. Respective length of growth compared with the control ± standard deviation (* *p* < 0.05, ** *p* < 0.01 vs. control, *n* = 9).

**Figure 6 antibiotics-12-00596-f006:**
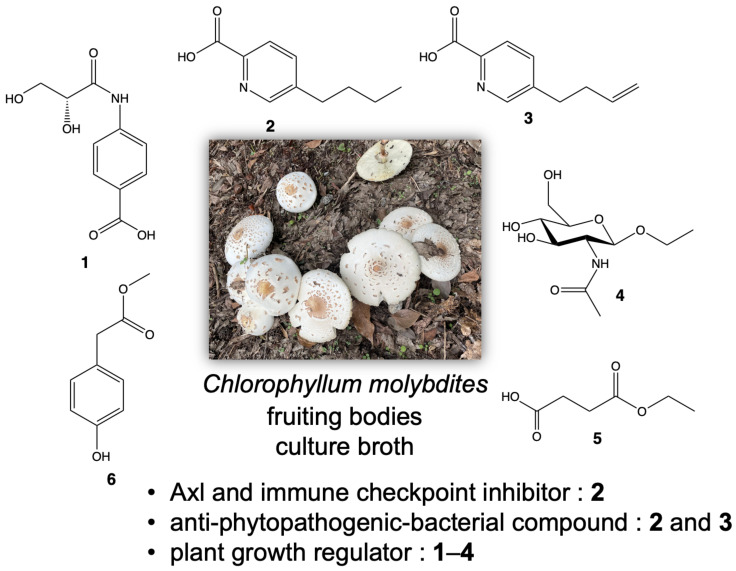
Summary of this study.

**Table 1 antibiotics-12-00596-t001:** ^1^H and ^13^C NMR Data for **1** in CD_3_OD.

Position	*δ*_H_ (*J* inHz)	*δ*c
1-COOH	−	171.0
1	−	129.5
2, 6	7.96 (d, 8.4)	131.6
3, 5	7.71 (d, 8.4)	120.4
4	−	142.7
1′	–	173.6
2′	4.20 (dd, 4.6, 3.7)	74.6
3′	3.80 (dd, 11.6, 4.6) 3.83 (dd, 11.6, 3.7)	65.3

## Data Availability

Not applicable.

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
