# Peer review of "Bioactive Compounds from the Mushroom-Forming Fungus Chlorophyllum molybdites"

_antibiotics, 2023, doi:10.3390/antibiotics12030596_

Round 1

Reviewer 1 Report

As a reviewer, I am very honored to see the application of mushroom compounds in human cancer diseases and plant disease resistance. Seeking advantages and avoiding disadvantages has always been the direction of our research. The fungal compounds could give us a better idea of where natural products might be heading in future research. Here, there are some minor question for author to response:

1.     Line 175-183: How does the presentation of method here relate to the bar graph and electrophoretic diagrams in Figure 3? If so, are these bar graphs the gene expression of cancer-related immune genes? Is the electrophoretic image a western blot? It is very difficult for us to believe that compound 2 is really related to the immune system by using RNA as a running gel to illustrate the gene expression. Please explain clearly. I suggest that gene expression is a better indicator.

2.     Line 221: “Chlorophyllum molybdites” should be written as “C. molybdites”,

3.     Line 295: 2,4-dichlorophenoxyacetic acid (2,4-D), Abbreviations should be added. 

4.     In the table 1, the “d" should be in italics.

5.     Figure 5, The concentration of compounds 2, 3 is wrong, I recommend that all figures are differentiated, for example by using A, B and C.

6.     Citation format needs to be carefully checked.

Author Response

Comments of Reviewer #1

Comments and Suggestions for Authors:

 As a reviewer, I am very honored to see the application of mushroom compounds in human cancer diseases and plant disease resistance. Seeking advantages and avoiding disadvantages has always been the direction of our research. The fungal compounds could give us a better idea of where natural products might be heading in future research. Here, there are some minor questions for author to response:

  1. Line 175-183: How does the presentation of method here relate to the bar graph and electrophoretic diagrams in Figure 3? If so, are these bar graphs the gene expression of cancer-related immune genes? Is the electrophoretic image a western blot? It is very difficult for us to believe that compound 2 is really related to the immune system by using RNA as a running gel to illustrate the gene expression. Please explain clearly. I suggest that gene expression is a better indicator.

Response: Thank you for your review of our manuscript and patient guidance. The bar graph and electrophoretic diagrams are the same result. The electrophoretic diagrams: PCR products were electrophoresed on a 1.5% agarose gel and then stained with ethidium bromide solution. The bar graph: Semi-quantitative RT-PCR results were quantified by using ImageJ software (Lines 217–219, references 41­–43, 63).

  1. Line 221: “Chlorophyllum molybdites” should be written as “C. molybdites”.

Response: It has been corrected as suggested (line 247).

  1. Line 295: 2,4-dichlorophenoxyacetic acid (2,4-D), Abbreviations should be added.

Response: It has been corrected as suggested (line 330).

  1. In the table 1, the “d" should be in italics.

Response: It has been corrected as suggested (Table 1).

  1. Figure 5, The concentration of compounds 2, 3 is wrong, I recommend that all figures are differentiated, for example by using A, B and C.

Response: It has been corrected as suggested (Figure 5).

Reviewer 2 Report

Manuscripts are well written and English is well established. I suggest adding one more image, a kind of graphical abstract, but placed in the discussion. This will make it easier to understand the contents of the paper and increase readability.

I hope authors will do that.

Author Response

Thank you for your kind suggestion. We have seriously considered your suggestion and added Figure 6 in the manuscript.

Reviewer 3 Report

1- the introduction must be start about the mushroom fungus then its application on cancer tratments and other biological activities

2-  in the introduction : In addition, there are many re- ports demonstrating the beneficial effects of mushrooms on cancer treatment [15]. pleaase site another references 

3- in the introduction :Plant diseases are one of the major causes of global total crop production losses. The 55 severity of disease outbreaks caused by plant phytopathogenic fungi and bacteria has 56 been steadily increasing over the past decades [19]. please add another references such as :Australian Journal of Basic and Applied Sciences, 3(3): 1527-1539, 2009     Efficacy of fungal Rust Disease on Willow Plant in Egypt 

fungal leaf spot of maize: pathogen isolation, identification and host biochemical characterization TM Ghany Mycopath 10 (2

4- why not test the cytotoxicity of isolated compounds

5-  is  fusaric acid produced from mushroom 

6- in the antibacterial activity  why not used the solvent of extraction as  negative control

Author Response

Comments and Suggestions for Authors:

  • the introduction must be start about the mushroom fungus then its application on cancer treatments and other biological activities

Response: Thank you for your kind suggestion. We sincerely accepted your suggestion and made changes in the introduction of the manuscript. (Lines 39~72).

  • in the introduction : In addition, there are many reports demonstrating the beneficial effects of mushrooms on cancer treatment [15]. pleaase site another references

Response: As your suggested, references [16–20] have been citated in the introduction. (Lines 65~72).

  • in the introduction : Plant diseases are one of the major causes of global total crop production losses. The severity of disease outbreaks caused by plant phytopathogenic fungi and bacteria has been steadily increasing over the past decades [19]. please add another references such as :Australian Journal of Basic and Applied Sciences, 3(3): 1527-1539, 2009 Efficacy of fungal Rust Disease on Willow Plant in Egypt. fungal leaf spot of maize: pathogen isolation, identification and host biochemical characterization TM Ghany Mycopath (2012).

Response: As your suggested, references [45, 46] have been citated in the introduction. (Lines 102–104).

  • why not test the cytotoxicity of isolated compounds

Response: A protein molybdophyllysin in this mushroom are known as toxic [reference 25]. In this study, we attempted to find other active compounds from this mushroom.

  • is fusaric acid produced from mushroom

Response: Fusaric acid has been isolated from the culture filtrate of Fusarium nygamai. In this study, the compound was isolated from the culture broth of this fungus. Therefore, Fusaric acid was first isolate from mushroom-forming fungi (Line 292).

6- in the antibacterial activity why not used the solvent of extraction as negative control

Response: A total of 40 mL of MeOH were put on a paper disc (8 mm in diameter). After the discs were dried in the air, they were put on the medium as negative control (lines 232–234). Since the solvent was removed, so we do not think that the solvent of extraction as negative control was necessary.

Round 2

Reviewer 3 Report

line 47-51  need more cite with references 

Soliman, A.; Abdelbary, S.; Yonus, A.; Abdelghany, T. Trends in assessment of Ganoderma lucidum methanol extract against MRSA infection in vitro and in vivo with nutrition support. J. Adv. Pharm. Res. 20226, 46–57

Al-Rajhi, A. M. H., Alawlaqi, M. M.,  Abdel Ghany, T. M., and Moawad, H. (2023). “Amanita sp. from subtropical region of Saudi Arabia as a source of chitinase enzyme and its antifungal activity,” BioResources 18(2), 2928-2939. 

Author Response

line 47-51  need more cite with references

Response: Two references [4,5] have been added.

Soliman, A.; Abdelbary, S.; Yonus, A.; Abdelghany, T. Trends in assessment of Ganoderma lucidum methanol extract against MRSA infection in vitro and in vivo with nutrition support. J. Adv. Pharm. Res. 2022, 6, 46–57Al-Rajhi, A. M. H., Alawlaqi, M. M.,  Abdel Ghany, T. M., and Moawad, H. (2023). “Amanita sp. from subtropical region of Saudi Arabia as a source of chitinase enzyme and its antifungal activity,” BioResources 18(2), 2928-2939.

Response:  The two references [7,8] have been added.